# The “FIFTY SHADOWS” of the RALES Trial: Lessons about the Potential Risk of Dietary Potassium Supplementation in Patients with Chronic Kidney Disease

**DOI:** 10.3390/jcm11143970

**Published:** 2022-07-08

**Authors:** Gregorio Romero-González, Jordi Bover, Javier Arrieta, Davide Salera, Maribel Troya, Fredzzia Graterol, Pablo Ureña-Torres, Mario Cozzolino, Luca Di Lullo, Pietro E. Cippà, Marina Urrutia, Javier Paúl-Martinez, Ramón Boixeda, José Luis Górriz, Jordi Ara, Antoni Bayés-Genís, Antonio Bellasi, Claudio Ronco

**Affiliations:** 1Nephrology Department, University Hospital Germans Trias i Pujol (HGiTP), 08916 Badalona, Spain; iatros36@icloud.com (G.R.-G.); mitroya.germanstrias@gencat.cat (M.T.); fagraterol.germanstrias@gencat.cat (F.G.); m.urritia.jou@gmail.com (M.U.); javierpaulmartinez@gmail.com (J.P.-M.); jaradelrey@gmail.com (J.A.); 2REMAR-IGTP Group, Germans Trias i Pujol Research Institute (IGTP), Can Ruti Campus, 08916 Barcelona, Spain; 3International Renal Research Institute of Vicenza, 36100 Vicenza, Italy; cronco@goldnet.it; 4Nephrology Department, University Hospital Basurto, 48013 Bilbao, Spain; jarrietal@senefro.org; 5Department of Medicine, Division of Nephrology, Ente Ospedaliero Cantonale, 6900 Lugano, Switzerland; davide.salera@eoc.ch (D.S.); pietro.cippa@eoc.ch (P.E.C.); antonio.bellasi@eoc.ch (A.B.); 6AURA Nord Saint Ouen Dialysis Service, 93400 Saint Ouen, France; pablo.urena@wanadoo.fr; 7Service d’Explorations Fonctionnelles Rénales, Hôpital Necker, Université Paris V, René Descartes, 75006 Paris, France; 8Renal Division, ASST Santi Paolo e Carlo, Department of Health Sciences, University of Milan, 20122 Milan, Italy; mario.cozzolino@unimi.it; 9Nephrology Department, Parodi-Delfino Hospital, 00034 Colleferro, Italy; dilulloluca69@gmail.com; 10Internal Medicine Department, Mataró Hospital, 08304 Mataró, Spain; rboixedaviu@gmail.com; 11Department of Nephrology, Clínico University Hospital, INCLIVA, Universitat de València, 46010 Valencia, Spain; jlgorriz@gmail.com; 12Heart Failure Clinic and Cardiology Service, University Hospital Germans Trias i Pujol (HGTiP), 08916 Badalona, Spain; abayesgenis@gmail.com; 13CIBERCV, Instituto de Salud Carlos III, 28029 Madrid, Spain; 14Department of Nephrology, DIMED–University of Padova, 35122 Padova, Italy

**Keywords:** K^+^, hyperkalaemia, renin–angiotensin system, chronic kidney disease, salt substitutes, SSaSS, RALES

## Abstract

Hyperkalaemia (HK) is one of the most common electrolyte disorders and a frequent reason for nephrological consultations. High serum potassium (K^+^) levels are associated with elevated morbidity and mortality, mainly due to life-threatening arrhythmias. In the majority of cases, HK is associated with chronic kidney disease (CKD), or with the use of renin–angiotensin–aldosterone system inhibitors (RAASis) and/or mineral corticoid antagonists (MRAs). These drugs represent the mainstays of treatment in CKD, HF, diabetes, hypertension, and even glomerular diseases, in consideration of their beneficial effect on hard outcomes related to cardiovascular events and CKD progression. However, experiences in relation to the Randomised Aldactone Evaluation Study (RALES) cast a long shadow that extends to the present day, since the increased risk for HK remains a major concern. In this article, we summarise the physiology of K^+^ homeostasis, and we review the effects of dietary K^+^ on blood pressure and cardiovascular risk in the general population and in patients with early CKD, who are often not aware of this disease. We conclude with a note of caution regarding the recent publication of the SSaSS trial and the use of salt substitutes, particularly in patients with a limited capacity to increase K^+^ secretion in response to an exogenous load, particularly in the context of “occult” CKD, HF, and in patients taking RAASis and/or MRAs.

## 1. Introduction

How many nights have nephrologists spent in the hospital to dialyse patients with life-threatening hyperkalaemia (HK)? The answer is many, since HK is one of the most common electrolyte disorders and a frequent reason for nephrological consultation. High serum potassium (K^+^) levels are associated with elevated morbidity and mortality, mainly due to life-threatening arrhythmias [1,2,3,4]. Factors or comorbidities such as diabetes mellitus, heart failure (HF), age, metabolic acidosis, and high protein intake can be related to HK; however, chronic kidney disease (CKD) is by far the most important and frequent association, especially in patients using renin–angiotensin–aldosterone system inhibitors (RAASi) [3,4]. Some of these comorbidities and the use of RAASis tend to cluster in CKD, since RAASis improved hard outcomes (e.g., cardiovascular events and CKD progression) in several studies. New winds seem to be blowing for cardiorenal medicine. Novel concepts such as “congestive nephropathy” and “pseudoworsening” renal function [5,6] are emerging. Similarly, new drugs for the treatment of HF and new K^+^ binders are now available on the market, offering interesting new opportunities to individualise treatment. Indeed, hyperkalaemia prevention allows for the safer administration of life-saving drugs such as angiotensin-converting enzyme inhibitors, angiotensin-receptor blockers, and mineralocorticoid receptor antagonists (MRA) [7], which currently represent the mainstays of treatment in CKD, HF, diabetes, hypertension, and even glomerular diseases [8,9,10,11,12].

However, the publication of the seminal RALES trial [13] was not devoid of undesirable consequences [14]. The RALES trial was discontinued early after the demonstration of the clear beneficial effects of spironolactone in reducing the risk of death, decreasing the frequency of hospitalisation for HF, and improving symptoms as assessed using the New York Heart Association functional class, paradoxically with no evidence of severe HK in either group [13]. The prescription of spironolactone increased abruptly following the publication of the study, but this increase was accompanied by an increase in HK-associated morbidity and mortality among patients also treated with angiotensin-converting enzyme inhibitors [14]. In fact, there were several-fold increases in the number of hospitalisations for HK (requiring dialysis in some cases) and associated mortality (from 2.4 to 11.0/1000 patients and 0.3 to 2.0/1000 patients, respectively) [14]. Moreover, this HK risk extended for many years, as was reported even by primary care physicians [15]. These observations cast a long shadow that extends to the present day and raises some concerns about the real-world impact of some recent studies [14,15].

The interest in MRAs has recently been revived by the development of nonsteroidal MRAs such as finerenone, which reduce both cardiovascular events and the risk of progression of renal function in diabetic subjects with CKD [16]. An increasing number of patients are currently using RAASis and/or MRAs, in accordance with the various guidelines [8,9,10,11,12]. In parallel, the Salt Substitute and Stroke Study (SSaSS) trial [17] recently reported on the potential beneficial effects of increasing diet K^+^ content. However, caution should be exerted in the contest of an increasing number of people with chronic clinical conditions such as CKD, often neglected by the patient and often treated with RAASis [18]. In this context, “occult” renal insufficiency, represented by the presence of “normal” serum creatinine but decreased glomerular filtration rate (and/or pathological albuminuria), has frequently been neglected in studies [14,15,19]. Consequently, this article offers a note of caution regarding the recent publication of the SSaSS (Salt Substitute and Stroke Study) trial [17], particularly in patients with known or “occult” CKD, those with HF, and/or those taking RAASi.

## 2. The Complex Mechanisms Involved in Keeping Serum K^+^ under Control

In brief, total-body K^+^ load and the appropriate distribution of K^+^ across the cell membrane are vital for normal cellular function [20]. Total-body K^+^ load is mainly determined by changes in the excretion of K^+^ by the kidneys in response to intake levels [21]. Under normal conditions, kidney regulation is accompanied by the activation of neurohormonal mechanisms. Thus, insulin and catecholamines also make important contributions in maintaining the internal distribution of K^+^; however, despite these homeostatic pathways, disorders of K^+^ homeostasis are common, especially in CKD patients [21].

K^+^ is the most abundant intracellular cation; approximately 98–99% of total K^+^ is within the cells, and changes in concentration between the intracellular and extracellular compartments have important clinical consequences [20,21]. Thus, K^+^ levels need to be closely regulated to maintain plasma K^+^ concentration within a narrow range, matching K^+^ intake with excretion and ensuring a proper distribution between fluid compartments [20]. The slight difference between the intracellular and extracellular spaces determines the resting cellular voltage action potential (the threshold action potential is influenced by calcium, among other ions). To maintain a proper balance, there are both immediate and long-term mechanisms to regulate plasma K^+^ concentrations [21].

Cellular buffering is an immediate mechanism mediated by insulin, catecholamines (β-adrenergic tone), acid–base disorders, plasma tonicity, and plasma K^+^ itself, among others. Long-term renal mechanisms to avoid total-body K^+^ overloads involve all segments of the nephron. However, the most sophisticated regulation process occurs in the aldosterone-sensitive tubule (comprising the late distal convoluted tubule, the connecting tubule, and the cortical collecting duct) [21]. In these specialised areas of the nephron, several cotransporters are dedicated to the reabsorption of sodium (ENaC) and the secretion of K^+^ through ROMK and Max-K channels [20]. The electrical and chemical gradients determining the K^+^ flow into the urine are influenced by plasma K^+^ concentration, mineralocorticoid activity, distal sodium delivery, and tubular fluid flow rate (which is decreased in HF), among other factors [20].

HK is common in patients with acute kidney injury or advanced CKD (i.e., G4–G5), especially if they are under RAASi treatment (primary indications for which include not only CKD, but also diabetes and HF). As the adaptations that occur in the kidney (and gastrointestinal tract) in CKD patients have a limited capacity to increase K^+^ secretion further when there is an exogenous load, HK can occur following even a modest increase in K^+^ intake [20].

## 3. The SSaSS Trial

The results of the SSaSS trial were recently published in the New England Journal of Medicine [17]. This study comprised a total of 20,995 adult patients with a history of stroke (72.6%). The mean age was 65.4 years, and high blood pressure was present in 88.4%. The trial was conducted in 600 villages in rural China, with at least 35 participants per village and a mean follow-up of 4.74 years. The aim was to define the risks and benefits of using a salt substitute (75% NaCl and 25% KCl) compared with regular salt (100% NaCl). The authors observed that the use of the salt substitute significantly reduced the rates of stroke (RR: 0.86 (95% CI 0.77–0.96)), major cardiovascular events (RR: 0.87 (95% CI 0.80–0.94)) and death (RR: 0.88 (95% CI 0.82–0.95)), with no increased risk of serious adverse events attributable to HK (RR: 1.04 (95% CI 0.80–1.37); *p* = 0.76) [17]. Notably, subjects on a potassium-sparing diuretic or with serious renal impairment were excluded for the study. Additional advice to try to reduce the total amount of the salt substitute used compared to prior salt consumption was provided to the intervention group participants, questioning the general applicability of the study findings [17].

These results serve to broaden the interesting ongoing debate regarding the beneficial effects of increasing K^+^ content in the diet [22,23,24], even in CKD patients [25,26]; however, they also give rise to serious concerns on the basis of the initially negative previous experience following publication of the RALES study [13]. Moreover, there are concerns about the impact that such an important article may have on social media posts and global public health policies regarding the beneficial use of salt substitutes rich in K^+^. These concerns are of critical importance in patients with known or “occult” CKD and those taking RAASi, bearing in mind that the global prevalence of CKD is estimated to be around 9.1% [27], and that the prevalence may be much higher among the elderly (in whom the prevalence of “occult” CKD is raised, especially among women) [28] and patients with cardiovascular disease [29].

Lastly, the SSaSS trial has significant limitations in terms of the absence of information on renal function and the exclusion of potential participants due to “serious kidney impairment”. Moreover, the study lacked serial measurements of plasma K^+^; the number of participants with elevated K^+^ levels therefore remained essentially unknown, hindering extrapolating the results to the general population.

## 4. Dietary K^+^: Beneficial Effects on Blood Pressure and Cardiovascular Risk

The beneficial effects of K^+^ on blood pressure have been well-described in the literature [30,31,32]. The blood pressure-lowering effects of K^+^ are more pronounced in individuals who consume a high-sodium diet, suggesting that K^+^ influences salt sensitivity (K^+^ induces natriuresis) [33,34,35,36]. Studies have shown that K^+^ depletion directly activates renin, angiotensin II, and endothelin-1 in the kidney independently of the RAAS system [35]. Additionally, K^+^ may exert positive effects by decreasing the activity of the kidney’s sympathetic nervous system (attenuating the activity (increasing the turnover) of catecholamines). It has direct effects on vascular tone, mediated via endothelium-dependent vasodilation, and also increases endothelial nitric oxide activity, which in turn decreases arterial stiffness [34,37]. Accordingly, Oberleithner et al. showed that high extracellular K^+^ significantly reduces the stiffness of vascular endothelial cells by changing the endothelial cell structure and increasing the release of nitric oxide. In contrast, high extracellular sodium and aldosterone prevent these changes [37].

Regarding cardiovascular disease, the diet of our ancestors consisted mainly of vegetables, fruits, and game, providing small amounts of sodium and large amounts of K^+^ [35]. This diet is entirely different from current diets (in fact, it is essentially the opposite), which may at least partially explain the high prevalence of hypertension, cardiovascular disease, and CKD in the general population. Consistently, several studies found a direct association between higher dietary K^+^ intake and a lower incidence of cardiovascular disease [34,36,38], and the relationship of high K^+^ intake to better kidney outcomes may be even more pronounced [21]. Furthermore, a high K^+^ diet may lead to blood-pressure-independent protective effects such as anti-inflammatory, antifibrotic, and antioxidant effects, the improvement of endothelial function, and the prevention of atherosclerosis [39]. Thus, the beneficial effects of increasing dietary K^+^ intake on blood pressure and renal/cardiovascular outcomes are becoming increasingly evident from epidemiological, clinical, and experimental studies (as recently reviewed by Wei et al.) [35]. As another example, Araki et al. demonstrated that urinary K^+^ excretion >1.72 g/day (44 mmol/day) in patients with type 2 diabetes mellitus decreased the incidence of renal failure or cardiovascular events and at least halved the progression to stage 4 CKD [40].

Nevertheless, the beneficial effects attributed to K^+^ may not all be a direct consequence of increasing dietary K^+^; in some cases, they may result from a decrease in sodium intake [41]. This may be particularly relevant in regions and countries where sodium intake is high, such as in northern China, where the average daily intake is 11,200 mg (487 mmol) [42]. Indeed, the SSaSS authors proposed extending the use of salt substitutes in regions where salt consumption is high, such as Latin America, Asia, and Africa. In other words, as what happened after earlier positive studies when the World Health Organisation and the Institute of Medicine released new recommendations for increasing dietary K^+^ intake (of at least 3.5 and 4.6 g/day (90 and 120 mmol/day), respectively) [43,44], the SsaSS study may lead to the design of policies to promote salt substitutes by reforming products and processing these substitutes on a large scale. However, evidence regarding the positive effects of dietary K^+^ or salt substitutes (not equivalent) in patients with CKD (as well as in diabetic patients or patients with HF) is far from consistent.

## 5. Dietary K^+^ and CKD: Does Size (CKD Stage or K^+^ Load) Matter?

The close relationship among blood pressure, cardiovascular disease, and CKD has led to the hypothesis that dietary K^+^ may also protect the kidney, and that patients with CKD may benefit from increasing dietary K^+^ [35]. Indeed, such effects are supported by emerging evidence from epidemiological studies in humans and experimental data from animal models [35]. Although these reports and cohorts, such as that included in the SSaSS trial, suggest that higher K^+^ intake may be protective for cardiovascular health in the general population, a low K^+^ intake is often recommended in patients with decreased renal function and/or those with decreased RAAS activity due to the risk of HK and the fear of adverse serious events (Figure 1). Indeed, as the glomerular filtration rate decreases, K^+^ levels tend to increase, and the prevalence of plasma K^+^ > 5.0 mmol/L is around 12–18% in CKD patients compared with 3–4% in patients without CKD [45,46,47]. While observational data suggest that even milder elevations of K^+^ levels above 4.5 mmol/L are associated with a higher risk of mortality [45], some other lines of evidence suggest a U-shaped association of K^+^ levels and a risk of unfavourable outcomes in CKD patients, with the lowest risk observed with K^+^ between 4.6 and 5.3 mEq/L in haemodialysis (HD) patients [48,49]. Hence, more “permissive” K^+^ levels may be allowed in these patients. Furthermore, despite the assumption that high dietary K^+^ may predispose to HK, clear supportive evidence is lacking [39], and awaits confirmation in properly designed studies in CKD and HD patients [50].

A recent study investigated in a cross-sectional analysis whether dietary K^+^ or the consumption of certain food groups related to K^+^ intake was associated with HK in a non-dialysis-dependent CKD (NDD-CKD) cohort and an HD cohort [39]. Dietary K^+^ intake was assessed by means of 3-day food records. The NDD-CKD cohort included 95 patients with an estimated median glomerular filtration rate of 23 mL/min/1.73 m^2^ (CKD G4), and the HD cohort (CKD G5D) included 117 patients. Somewhat surprisingly, the authors did not find an association between serum K^+^ and either dietary K^+^ or the consumption of selected food groups in NDD-CKD patients with HK (36.8%). Conditions associated with HK in multivariable analysis were diabetes mellitus and metabolic acidosis. Similarly, no association was found between serum K^+^ and either dietary K^+^ or the consumption of selected food groups in HD patients with HK (50.5%). The authors consequently concluded that dietary K^+^ is not associated with serum K^+^ or HK in either NDD-CKD or HD patients, and that, before restricting dietary K^+^, other potential clinical factors related to serum K^+^ balance should be considered. They also acknowledged significant limitations regarding the study’s ability to establish a cause–effect relationship: there may have been errors in assessing K^+^ intake and an inability to differentiate the sources of K^+^ in the diet [51]. Another large observational study of 8043 subjects receiving maintenance dialysis also questioned the axiom of high K^+^ intake being associated with higher risk of HK and death [50].

Most observational studies have demonstrated that diets rich in K^+^ are associated with better kidney outcomes in the overall population or in patients with early CKD [35]. However, a recent systematic review failed to conclusively demonstrate the effects of dietary K^+^ intake on CKD progression, possibly due to the study heterogeneity, the relatively low range of overall dietary K^+^ intake reported in different studies (average intake below the 2004 Kidney Disease Outcomes Quality Initiatives guidelines [52]), and concerns about the methods used to assess K^+^ intake in most studies (i.e., kaliuria) in late stages of CKD [53]. Intriguingly, patients undergoing peritoneal dialysis, which removes small amounts of K^+^, seem to absorb less K^+^ from their diets [54].

A multidisciplinary group of researchers and clinicians met in October 2018 to identify evidence and address controversies relating to K^+^ management in CKD, and this group recently published the conclusions of their Kidney Disease Global Outcomes (KDIGO) controversy conference [21]. The authors underlined that there is “increasing evidence showing beneficial associations with plant-based diet and data to suggest a paradigm shift from the idea of dietary restriction toward fostering patterns of eating that are associated with better outcomes”, and a “paucity of data on the effect of dietary modification in restoring abnormal plasma K^+^ to the normal range” [55,56]. They also offered guidance on the evaluation and management of dyskalaemias in the context of kidney diseases and research priorities. Consequently, studies have been conducted in which patients with progressive CKD (i.e., G3b or G4) have been treated with different K^+^ supplements, with analysis of the change in estimated glomerular filtration rate, and secondary outcomes such as 24 h blood pressure and albuminuria [57]. These intervention studies may be expected to contribute more robust clinical evidence to support the recent exciting insights into the physiology and epidemiology of K^+^ homeostasis, and to cast light on the risk/benefit ratio of treatment with K^+^ supplements in CKD.

Future studies should also evaluate nutritional intervention in light of the concomitant use of drugs that modulate the RAASi, such as ACE-I, ARBs, MRA, and/or sodium-glucose transporter 2 inhibitors (SGLT2i), beta blockers, or diuretics. Indeed, recent evidence supports the use of finerenone on top of maximal ACEi/ARB tolerated doses in diabetic patients to reduce the risk of both cardiovascular and renal outcomes at the cost of a small but statistically significant increase in serum K^+^ levels. Indeed, as documented in a recent pooled analysis of 13,171 patients with diabetes mellitus and CKD G3 and G4 enrolled in the FIGARO-DKD and FIDELIO-DKD studies, the serum levels of K^+^ increased by 0.2 mmol/L among patients treated with finerenone [16]. As compared to a placebo, a twofold (HR 2.13; 95%CI: 1.86–2.45) increase in the risk of HK (defined as serum K^+^ > 5.5 mmol/L) when finerenone was added to maximal dose of ARB/ACEi was observed [16]. In this context, however, the use of diuretics, SGLT2i, or betablockers could also modulate serum K^+^ levels, casting further questions on the optimal therapeutic strategy in a specific subgroups of patients to ameliorate renal and cardiovascular outcomes, and prevent the occurrence of HK.

## 6. Dietary K^+^ and CKD: A Word of Caution

Several important gaps in knowledge remain even after the publication of the SSaSS trial, and, as previously mentioned, a note of caution is advised. First, previous association studies do not prove causality. Second, the effects of lower sodium cannot be separated from those of the higher K^+^ delivered by the salt substitute in this study. Third, the bioavailability of inorganic K^+^ (as provided in the SSaSS trial) is not necessarily equivalent to that of K^+^ derived from a fruit- or vegetable-rich diet, as we already know from the strikingly different intestinal bioavailability of phosphate from inorganic (i.e., additives) versus plant-derived phosphate sources [19,58]. The anion linked to K^+^, other nutrients (e.g., magnesium, vitamin K), protein (with lower phosphate bioavailability), higher fiber content (enhancing intestinal motility and short-chain fatty acid production), or alkali (neutralising acidosis and its harmful consequences) in plant-based or adapted “healthy” diets may have contributed to the outcomes of different studies [25,59,60,61]. These factors may also explain the unclear correlation between dietary and serum K^+^ [51,61]. As was recently reported [25,61], the traditional dietary paradigm limiting the intake of fruits and vegetables to CKD patients because of their high K^+^ content is rapidly evolving due to the potential pleiotropic benefits deriving from a fundamentally vegetarian or Mediterranean-adapted diet [59,60]. These also include improvement in gut dysbiosis and decreased production of harmful uraemic toxins, together with reductions in inflammation and oxidative stress [25]. Lastly, although HK is the main concern with these diets, good cooking techniques can minimise the amount of absorbed K^+^ [25]; however, concomitant medications should be considered when K^+^ enriched salts or diets are suggested in light of the potential interaction with impaired renal function and/or excretion and the risk of HK.

Lastly, studies have shown that K^+^ supplementation increases the tubular phosphate reabsorption capacity in rats and increases serum phosphate in healthy individuals [44,62], indicating a possible interaction among K^+^, phosphate, and fibroblast growth factor-23 [35]. Both phosphate and fibroblast growth factor 23 are causes of cardiovascular mortality in CKD [63]. Thus, nephrologists used to say that K^+^ kills rapidly, whereas phosphate kills slowly [64,65].

## 7. Conclusions

Globally, 850 million people have CKD, and it is projected that by 2040, it will be the fifth leading cause of death in the world [66]. Moreover, cardiovascular disease is more frequent and more severe in CKD patients than in the non-CKD population, primarily due to nonatherosclerotic pathologies such as HF, in which treatments with RAASis and/or MRAs are the cornerstone [8,9,10,11,12]. Nevertheless, HK represents a significant concern and limitation.

Despite advances in our knowledge of the potential benefits of increasing K^+^ intake (via vegetable-based diets or salt substitutes), the evidence is still blurred and scarce, especially in elderly CKD patients and patients with HF who are taking RAASis and/or MRAs. As explained above, adaptations that occur in the kidneys and gastrointestinal tract in CKD patients result in a limited capacity to increase K^+^ secretion when there is an exogenous load or inhibition of the RAAS system.

Fortunately, the development of new RAASis and/or MRAs such as finerenone [67,68], and new K^+^ binders with better safety profiles and tolerability, such as patiromer or sodium zirconium cyclosilicate [69], may mitigate the risk of HK and allow for life-saving drugs to be administered rather than withdrawn [56]. Moreover, cardio- and nephroprotective drugs such as sodium-glucose cotransporter 2 inhibitors may also reduce the risk of HK [70]. Nevertheless, recommending salt substitutes indiscriminately on the basis of the SSaSS study, without knowing the basal dietary K^+^ content and without proper K^+^ and/or renal function monitoring, may extend further the “long shadow” of the RALES trial. Broadcasting and encouraging global public health policies on the beneficial use of salt substitutes rich in K^+^ without appropriate warnings may not be desirable. In this regard, food manufacturers are increasingly substituting KCl in food products so as to reduce the sodium chloride content [71], and warnings have indeed been issued about the serious and potentially fatal consequences for people who need to restrict dietary K^+^ [71], such as those to which this review refers.

## Figures and Tables

**Figure 1 jcm-11-03970-f001:**
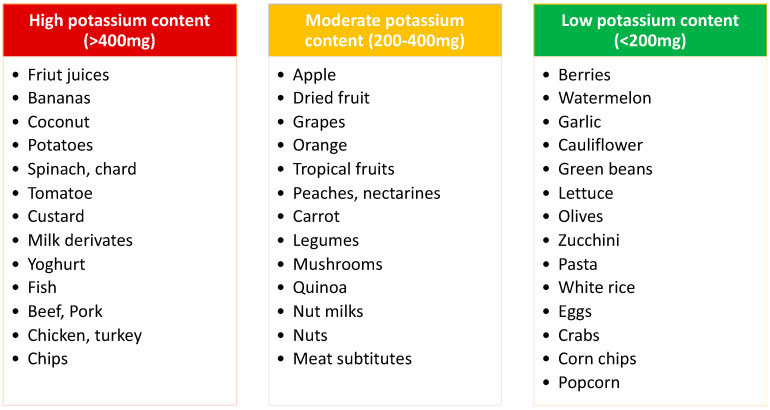
Dietary potassium content. Adapted from [21].

## Data Availability

Not applicable.

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
