# Peer review of "The “FIFTY SHADOWS” of the RALES Trial: Lessons about the Potential Risk of Dietary Potassium Supplementation in Patients with Chronic Kidney Disease"

_jcm, 2022, doi:10.3390/jcm11143970_

Round 1

Reviewer 1 Report

Overall a very interesting and evidence-based discussion of an important topic that should inform a very important discussion of an issue with significant morbidity and mortality implications. 

Recommend

1.  Some discussion on Page 3 in paragraph lines 138-147 as to whether the authors of the SSaSS trial (line 129 citation should be 20 and. not 14) described the K and Na and animal source protein content of the diet.  Notably diet low in calcium.  Would be particularly important when trying to apply the results of this study to other populations.  This is a question that is raised later in the discussion.

2.  This reviewer is not aware of any studies that clearly delineate the bioavailability of K with a mostly vegetarian diet versus a meat based diet etc., in a manner analogous to the studies on phosphorus but if they are available would add this to the discussion.

3.  Minor recommendation:  Some more careful proofing and copy editing. 

Some Examples: 

a). Long phrase lines 60-63; second half of sentence lacks a verb. 

b). Sentence 84-86 is not grammatically correct. Needs a verb and should this be "increasing number of people" and also "in this context of an epidemiological context" 

c). Reference line 96; 15-16 superscript, and line 129 reference 20

Mostly reads very well and raises a provocative question.  Should prompt some more careful investigations and maybe proper robust measurements during future clinical trials.  

Author Response

Dear Editor,

thank you for giving us the opportunity to respond to the reviewers. In general, we think the paper has improved thanks to the reviewers' comments and suggestions. We enclose a point-by-point response to the reviewers and a new version of the paper with the changes highlighted in yellow.

We hope that this new version can be considered acceptable for publication and we remain available for any further clarification if required.

Warm regards

On behalf of the authors.

Antonio Bellasi

Here below, please find the point-by-point reply to the reviewer’s request

Recommend

Some discussion on Page 3 in paragraph lines 138-147 as to whether the authors of the SSaSS trial (line 129 citation should be 20 and. not 14) described the K and Na and animal source protein content of the diet.  Notably diet low in calcium.  Would be particularly important when trying to apply the results of this study to other populations.  This is a question that is raised later in the discussion.

Answer: we thank the reviewer for the suggestion. We have added a comment to the discussion and edited the citation (highlighted in yellow)

This reviewer is not aware of any studies that clearly delineate the bioavailability of K with a mostly vegetarian diet versus a meat based diet etc., in a manner analogous to the studies on phosphorus but if they are available would add this to the discussion.

Answer: we thank the reviewer for this interesting question. We are not aware of studies that have investigated potassium bioavailability of different food sources. Although it is plausible that potassium bioavailability is influenced by different nutrients, it has also to be noted that potassium is found in most plant and animal tissues, with fruits and vegetables having a higher potassium density than cereals and animal foods. However, contrary to phosphorous, potassium is intrinsically soluble and quickly dispersed in the luminal water of the upper digestive tract making it in general more absorbable than phosphorous. Little is known about the bioavailability of potassium, with the majority of work being centered on the assessment of urinary potassium losses after potassium intake.

Minor recommendation:  Some more careful proofing and copy editing. 

Some Examples: 

a). Long phrase lines 60-63; second half of sentence lacks a verb. 

Answer: we thank the reviewer for pointing this out. The sentence has now been edited highlighted in yellow)

b). Sentence 84-86 is not grammatically correct. Needs a verb and should this be "increasing number of people" and also "in this context of an epidemiological context" 

Answer: we thank the reviewer for the remark. The sentence has now been edited (highlighted in yellow)

c). Reference line 96; 15-16 superscript, and line 129 reference 20

Answer: we thank the reviewer for the remark. The reference style and number have now been edited highlighted in yellow).

Mostly reads very well and raises a provocative question.  Should prompt some more careful investigations and maybe proper robust measurements during future clinical trials

Answer: we thank the reviewer for the praise. This is the aim of the review article.

Reviewer 2 Report

Authors focused on potassium dietary supplementation and its significance both in general population and those suffering from chronic kidney disease. It is already known that lowering sodium intake and paralel increase of kalium might have multiple positive effects. On one hand, lowering sodium consumption is related to lowering blood pressure therefore having protective impact. On the other hand higher potassium levels may have positive influence but it should be bear in mind that CKD patients may be at risk especially when used together with drugs such ACEI or spironolactone. Therefore in my opinion the paper is worth of publishing and I have no further remarks.

Author Response

We thank the reviewer for the praise.